

# Exploring the use of mobile health among patients with cardiometabolic and respiratory chronic diseases in primary care nursing: a cross-sectional study

Daniel Monasor Ortola[1,2,3], José Joaquín Mira[1,4], Antonio Esteve Ríos[2,5,6] and Virginia García Ferrer[3]

[1] Health Psychology Department, Universidad Miguel Hernández de Elche, Elche, Alicante, Spain
[2] Department of nursing, Universidad de Alicante, San Vicente del Raspeig, Alicante, Spain
[3] Denia Health Department, Conselleria de Sanidad, Denia, Alicante, Spain
[4] Alicante-Sant Joan Health Department, Conselleria de Sanidad, Alicante, Alicante, Spain
[5] Alicante Institute for Health and Biomedical Research (ISABIAL), Alicante, Spain
[6] Group of Person-Centred Care and Innovation in Health Outcomes, Universidad de Alicante, San Vicente del Raspeig, Alicante, Spain

Corresponding author
Daniel Monasor Ortola,
monasor_dan@gva.es

## ABSTRACT

**Background:** Chronic diseases, such as cardiometabolic and respiratory conditions, are major contributors to global morbidity and mortality. These conditions place a considerable burden on primary care due to rising healthcare costs and increasing patient dependency in daily life. Digital tools, such as mobile health (mHealth), have emerged as promising resources to support chronic disease management in this setting. However, patient adoption of mHealth remains limited, and few studies have specifically examined its use among individuals with multiple chronic conditions. This study aims to examine the overall use and implementation of mHealth technologies among patients with chronic cardiometabolic and/or respiratory diseases receiving care from primary care nursing services.

**Methods:** A descriptive cross-sectional observational study was conducted from December 2022 to October 2023 in the primary care services of the Denia Health Department, Alicante, Spain. A non-probability purposive-consecutive sampling method was used. Participants were adult patients attending chronic care nursing consultations with one or more of the following diagnoses: hypertension, type 2 diabetes mellitus, chronic obstructive pulmonary disease, dyslipidemia, or obesity. Sociodemographic and clinical variables were assessed, and participants completed an *ad hoc* questionnaire on mobile phone use, health app usage, and online health information-seeking behaviors. Statistical analysis was performed using RStudio for macOS®. Descriptive statistics, chi-square tests, and t-tests were applied.

**Results:** A total of 523 participants were included (mean age: 71.65 years; SD = 11.91; range: 21–99 years; 50.9% male). Mobile phone and health app usage were significantly higher among men ($P = 0.0017$) and younger patients, who also demonstrated greater digital proficiency and experience ($P < 0.001$). Patients with diabetes were the most frequent users of these tools. Those willing to use mHealth for monitoring were also younger and more digitally skilled ($P < 0.001$). Most participants searched for health information online using search engines (81.8%),

and among those who consulted official sources, 69.2% had secondary or higher education.

**Conclusions:** The use of mHealth technologies in primary care is more common among younger patients, those with higher education levels, and individuals with greater digital skills, suggesting a digital divide. Patients with diabetes were the most frequent users. These findings highlight the importance of promoting digital literacy and designing accessible, user-friendly tools tailored to patients' individual needs. It is also crucial to consider the preference for face-to-face care, particularly among older adults. Effective implementation of digital health solutions requires inclusive strategies that account for technological, socioeconomic, and contextual differences.

## INTRODUCTION

Chronic diseases pose a global health challenge, causing 41 million deaths annually, accounting for 71% of all deaths. Of these, 21 million are due to cardiometabolic diseases, 13 million to cancer, and 4 million to chronic respiratory diseases (*World Health Organization, 2023a*). These conditions lead to progressive health deterioration, increased dependency, and high costs, particularly impacting primary care (PC) (*World Health Organization, 2022a*).

In fact, 80% of all PC consultations involve patients with chronic diseases, consuming 70% of the total health budget (*Conselleria de Sanitat, 2014*). Among the most common chronic diseases in adults over 15 years old are hypertension (19%), dyslipidemia (15.3%), osteoarthritis (14.4%), and type 2 diabetes mellitus (DM) (7.5%), with a higher incidence in economically disadvantaged populations and those with lower education levels (*Instituto Nacional de Estadística, 2018*, *2023a*; *World Health Organization, 2022b*). In this context, nursing consultations in PC play a key role in the monitoring, care, and education of patients with chronic cardiometabolic and respiratory conditions, aiming to maintain health and improve their quality of life (*Instituto Español de Investigación Enfermera, Consejo General de Enfermería de España, 2021*).

The "Seville Declaration", resulting from Spain's national conference on chronic diseases, established key measures to improve disease management and strengthen coordination with PC, including, for the first time, the evaluation of the use of information and communication technologies (ICTs) to optimize professional–patient interactions (*Orozco-Beltrán & Baturone, 2011*).

Today, organizations such as the World Health Organization (WHO) and the European Commission (EC) have developed digital strategies and initiatives to support the management of challenges such as chronicity, population aging, and data exchange for healthcare service delivery across the EC (*European Commission, 2025*; *World Health Organization, 2021*).

In response to these challenges, the healthcare system is evolving towards a patient-centered model, increasingly driven by ICTs, including telemedicine, telenursing,

and mobile health (mHealth) solutions (*González Esteban et al., 2016*; *Morales-Asencio, 2014*; *Tambo-Lizalde et al., 2021*; *World Health Organization, 2016*; *World Health Organization, 2018*).

Furthermore, emerging technologies like artificial intelligence (AI) and the Internet of Medical Things (IoMT) offer new opportunities to improve healthcare delivery and patient outcomes (*Alòs et al., 2024*; EU), especially following the COVID-19 pandemic, which acted as a catalyst for digital transformation in healthcare (*Almalki & Giannicchi, 2021*; *Gomes-de Almeida, Marabujo & do Carmo-Gonçalves, 2021*). Within this context, the WHO defines mHealth as the use of mobile phones, tablets, wireless monitoring devices, digital assistants, and related applications for diagnosis, monitoring, prevention, and rehabilitation (*Alcazar & Ambrosio, 2019*; *Mayer, Rodríguez Blanco & Torrejon, 2019*; *Mira, 2018*; *Pérez-Jover et al., 2019*; *World Health Organization, 2016*). mHealth applications are commonly classified by criticality: general information; primary prevention and health promotion; secondary and tertiary prevention, including patient support and therapeutic education and data analysis with treatment impact (*Haute Autorité de Santé (HAS), 2016*). In PC, mHealth has proven particularly useful for managing chronic cardiometabolic conditions, with nurses showing a high degree of perceived competence and acceptance toward its use (*Monasor-Ortola, Mira-Solves & Esteve-Ríos, 2025*).

In Spain, one example of mHealth implementation in PC is *ValCrònic*, an initiative in the Valencian Community (VC), Spain, under the Chronic patient care improvement plan. This pilot telemonitoring program, designed to optimize chronic disease management in PC, improved key health parameters and led to a 50.9% reduction in PC emergency visits and a 32.3% decrease in hospital admissions (*Mira-Solves et al., 2014*; *Monasor-Ortola, Mira-Solves & Esteve-Ríos, 2025*; *Orozco-Beltrán et al., 2017*).

Despite its potential, mHealth use in nursing has been largely limited to specialized units or focused on acute conditions and mental health which underscores the need to expand its use within community care (*Eriksson et al., 2020*; *Moore, Kelly & Melnyk, 2024*; *Yliluoma & Palonen, 2020*). However, few studies have examined the use of mHealth by nurses for patients with multiple chronic conditions in PC. Most existing research has focused on a single disease, such as DM or congestive heart failure, and typically includes patients under the age of 65 (*Shiyab et al., 2024*; *Sockolow, Buck & Shadmi, 2021*; *Zhang et al., 2024*). Furthermore, methodological heterogeneity has limited the ability to generalize findings and has posed challenges to validating mHealth as an effective intervention in this setting (*Pascual-de la Pisa et al., 2020*). Nevertheless, the potential impact of these technologies on health outcomes remains significant (*Alcazar & Ambrosio, 2019*; *World Health Organization, 2016*).

On the other hand, nurses do not usually recommend the use of mHealth to chronic patients, although they would be willing to do so—just like other members of the medical staff—given its relevance in PC for chronic disease management, particularly in areas such as promoting patient self-monitoring and self-management, improving clinical follow-up, and facilitating the digitalization of the electronic health record (*Instituto Español de Investigación Enfermera, Consejo General de Enfermería de España, 2021*; *Monasor-Ortola, Mira-Solves & Esteve-Ríos, 2025*; *Zhang et al., 2024*).

Moreover, the security and privacy of digital data represent another key factor in the development and adoption of mHealth technologies. The *European Commission (2025)* has established various legislative and technical frameworks to promote secure practices. In parallel, it is also essential to address the digital divide—a concept defined by the (*World Health Organization, 2021*) as the imbalance in access, use, skills, affordability, and effective utilization of information and communication technologies among different social groups or regions—as it remains a significant barrier to the equitable adoption of mHealth.

In light of these challenges and opportunities, the objective of this study was to examine the overall use and implementation of mHealth technologies among patients with chronic cardiometabolic and/or respiratory diseases who are under the care of PC nursing services.

# MATERIALS AND METHODS

## Study design

A cross-sectional descriptive observational study was conducted between December 2022 and October 2023 in the PC setting of the Denia Health Department (DHD), Spain.

## Participants

The study focused on residents of Marina Alta, VC, Spain, within the DHD, part of the VC Health Ministry. The DHD comprises Denia Hospital and 11 basic health zones (BHZs), which together include 50 healthcare facilities: 12 PC centers and 38 auxiliary clinics that operate under the coordination of the main centers. The system serves over 172,000 people, 42,000 of whom are over 65 years old. It is a publicly funded healthcare system that provides free access to medical services. In 2022, PC consultations totaled 720,156 in-person, 328,477 telephone, and 34,507 home visits, with 55.1% involving individuals over 60 (*Conselleria de Sanitat, 2022*).

A sample size of 466 participants was estimated (97% confidence interval, 5% margin of error). A non-probability purposive-consecutive sampling method was used. Participants were consecutively recruited during the study period by previously trained nurses from the 11 BHZs, who voluntarily agreed to participate without financial compensation. Following a coordination meeting held in each BHZ, a total of 40 nurses agreed to take part in the study. In each zone, the nursing supervisor was responsible for overseeing the process and ensuring proper implementation of the study, together with the principal investigators, who also supervised the progress of the work. Patients were recruited during their visits to PC nursing consultations for the follow-up of chronic conditions, whether attending a first-time or a follow-up appointment. Nurses invited patients who met the inclusion and exclusion criteria to participate in the study. If the patient agreed, informed consent was obtained, and the nurse completed the data collection form. Inclusion criteria: patients enrolled in a chronic nursing program in PC, diagnosed with cardiometabolic and/or respiratory disease (DM, hypertension (HTN), Chronic Obstructive Pulmonary Disease (COPD), dyslipidemia, or obesity), aged over 18, and residing in one of the 11 BHZs. Exclusion criteria: patients who were bedridden, institutionalized, or had language barriers or cognitive impairments that prevented full participation in the interview.

A data anonymization protocol was implemented, assigning each patient a unique code composed of their initials followed by the last three digits of their Healthcare Identification Number.

## Measures

To analyze the study variables, the PC nursing staff conducted a clinical assessment and a closed interview using an *ad hoc* form (Table S1). The form had two sections: sociodemographic data (age, gender, education level, health center, number of consultations) and clinical data (medications, height (m), weight (kg), body mass index (BMI), blood pressure, digital blood glucose, diagnoses, inclusion in a specialized program, degree of chronicity, and perception of health status). It also included 13 questions on mobile health technology and internet use, divided into four blocks:

*1. Use of mobile phones and applications:* Three questions assessed usage, frequency, and skill in handling mobile phones and apps.
*2. Use of mobile applications for health:* Seven questions evaluated app/device use for health, reasons for use, and whether adoption began due to COVID-19.
*3. Use of mobile applications for monitoring chronic diseases in PC:* Three questions explored patients' willingness to use an app for monitoring chronic diseases with the PC team and its potential applications.
*4. Searching for health information on the internet:* Two questions assessed whether patients frequently searched for health information about treatments, conditions, or healthcare online and where.

An *ad hoc* form was developed to collect key variables related to mHealth use in a real-world nursing care setting, as no validated questionnaires in Spanish were identified that aligned with the study objectives.

## Ethical considerations

The study was approved on 7 November 2022 by the Denia Health Department Research Commission, which oversees ethical compliance. All participants received information sheets and provided written informed consent. A risk assessment, based on the Helsinki Declaration (2007), deemed the study negligible risk. The study also followed STROBE guidelines for transparent and complete reporting of observational research.

## Statistical analysis

Statistical analysis was performed using RStudio for macOS®. A descriptive analysis of qualitative variables with frequency tables and percentages was conducted. For this exploration, Q-Q normal plots, skewness and kurtosis indices, and the Kolmogorov-Smirnov test for goodness of fit to normality were used. Additionally, a box plot was employed. Quantitative variables were described using standard measures of centrality and variability. The Students t-test and ANOVA were used to compare means of different groups of subjects. Fisher's Exact Test and the Chi-square test of independence were used for comparing two categorical variables.

## RESULTS

### Descriptive statistics

A total of 523 individuals participated in the study, all of whom met the stipulated inclusion and exclusion criteria. The sociodemographic and clinical variables of the participants are presented in Table 1.

### Use of mobile phones and applications (Table 2)

Statistically significant differences indicated that mobile phone use was more frequent among younger individuals and men ($P < 0.001$). Men predominated in the group with the longest usage history (>20 years), while women were more represented among more recent users (*e.g.*, 5-year users: 61.7% female). A strong association was also observed between the duration of mobile phone use and perceived digital skill, with proficiency increasing with longer use ($\chi^2 = 172.60$; $P < 0.001$). Significant relationships were also observed between skill in handling and using apps and the duration of mobile phone use, with skill increasing as usage time increased ($\chi^2 = 172.60$; $P < 0.001$).

### Use of mobile health applications

Significant relationships were found between the variables of sex (male $n = 89/148$, 60.1%; $P = 0.017$) and age (mean = 65; standard deviation (SD) 10.9; $P = 0.004$) with the use of health-related devices, with higher usage among men and younger individuals.

Regarding patients who have used apps before or due to the COVID-19 pandemic (Table 3), statistically significant differences were found concerning age, with older patients being more likely to have used apps before the pandemic (t = −2.99; $P = 0.004$). In relation to COVID-19, only 11.3% (59/523) of participants used a specific app related to the disease, with 50.8% (30/59) of these using it for information searches.

Concerning the participants' pathologies, 63.5% (94/148) of those using health apps had HTN, and 58.8% (87/148) had DM. Only statistically significant differences were found in DM, as no significant differences were observed between other pathologies and their management through apps. Specifically, 96.1% (25/26; $\chi^2 = 17.92$; $P < 0.001$) of those using apps for glycemic control had DM. Age differences were also noted, with younger individuals more likely to use the apps to manage their conditions.

### Use of mobile apps for chronic disease monitoring in PC

Of the 523 participants, 38.8% (203) indicated they would use a mobile app to monitor chronic diseases in PC. These individuals reported greater proficiency in app use (72.9%, 102/140; $\chi^2 = 135.12$; $P < 0.001$) and more frequent app usage (67.8%, 154/227; $\chi^2 = 215.53$; $P < 0.001$). They were also significantly younger than those unwilling to use such apps (t = −10.51; $P < 0.001$) (Table 4). Perceived utility focused on clinical monitoring by the PC team (74.4%, 151/203), appointment management (66%, 134/203), and faster contact with health centers (50.7%, 103/203).

Among participants who indicated they would not use a mobile app for chronic disease monitoring, 58.3% (186/319) preferred in-person consultations. This group was predominantly composed of older adults (mean age 74; SD = 11.9; $P < 0.001$) and men

Table 1 Sociodemographic data and clinical variables (*n* = 523).

| Characteristic | Values |
|---|---|
| Sex, *n* (%) | |
| Female | 257 (49.14) |
| Male | 266 (50.86) |
| Age (years), mean (SD) | 72.6 (11.9) |
| Education level, *n* (%) | |
| None | 138 (26.39) |
| Primary | 254 (48.57) |
| Secondary or higher | 121 (23.14) |
| *n* In-Person Consultations, mean (SD) | 2.4 (1.5) |
| *n* Telephone Consultations, mean (SD) | 1.1 (1.6) |
| Clinical Variables, mean (SD) | |
| *n* Medications | 6.3 (3.6) |
| Weight (Kg) | 76.7 (17.0) |
| Height (m) | 1.6 (0.1) |
| BMI[a] | 28.5 (5.4) |
| SBP[b]/DBP[c] (mmHg) | 133.2 (14.0)/78.2 (9.9) |
| Blood sugar (mg/dl) | 131.6 (40.2) |
| Diagnoses, *n* (%) | |
| DM[d] | 310 (59.27) |
| HTN[e] | 393 (75.14) |
| COPD[f] | 28 (5.35) |
| Dyslipidemia | 230 (43.98) |
| Obesity | 102 (19.50) |
| Level of chronicity, *n* (%) | |
| G0[g] | 57 (10.9) |
| G1[h] | 152 (29.06) |
| G2[i] | 177 (33.84) |
| G3[j] | 132 (25.24) |
| Health Status Assessment, *n* (%) | |
| Very good | 36 (6.88) |
| Good | 279 (53.35) |
| Regular | 167 (31.93) |
| Poor | 33 (6.31) |
| Very poor | 8 (1.53) |

Notes:
[a] BMI, Body Mass Index.
[b] SBP, Systolic Blood Pressure.
[c] DBP, Diastolic Blood Pressure.
[d] DM, Diabetes Mellitus.
[e] HTN, Hypertension.
[f] COPD, Chronic Obstructive Pulmonary Disease.
[g] G0, Healthy individuals or acute problems.
[h] G1, Patient with risk factors.
[i] G2, Patient with moderate chronic complexity.
[j] G3, Patient with high chronic complexity or Palliative care.

**Table 2 Sex, age, skill, and app usage according to years of mobile phone use (n = 523).**

| Characteristic | Years of mobile phone use | | | | | P value |
|---|---|---|---|---|---|---|
| | Do not use (n = 49) | >5 years (n = 60) | >10 years (n = 125) | >15 years (n = 93) | >20 years (n = 195) | |
| **Age (years)** | | | | | | |
| Mean (SD) | 85.1 (8.6) | 76.5 (11.2) | 74.8 (10.5) | 69.6 (11.8) | 68.5 (10.9) | <0.001 |
| Median (IQR) | 85.0 (12.0) | 78.0 (12.5) | 76.0 (12.0) | 70.5 (15.5) | 68.0 (16.0) | |
| Range | 55.0–99.0 | 21.0–91.0 | 21.0–95.0 | 30.0–90.0 | 39.0–98.0 | |
| Missing, n (%) | 1 (2.04) | 1 (1.67) | | 1 (1.08) | 2 (1.03) | |
| **Sex, n (%)** | | | | | | |
| Male | 22 (44.90) | 23 (38.33) | 52 (41.60) | 36 (38.71) | 132 (67.69) | <0.001 |
| Female | 27 (55.10) | 37 (61.67) | 73 (58.40) | 57 (61.29) | 63 (32.31) | |
| **Do you consider yourself skilled with the mobile phone?, n (%)** | | | | | | |
| No | 46 (93.88) | 38 (63.33) | 73 (58.40) | 33 (35.48) | 66 (33.85) | <0.001 |
| Regular | 2 (4.08) | 18 (30.00) | 30 (24.00) | 24 (25.81) | 52 (26.67) | |
| Yes | 1 (2.04) | 4 (6.67) | 22 (17.60) | 36 (38.71) | 77 (39.49) | |
| **Do you use apps daily?, n (%)** | | | | | | |
| No | 47 (95.92) | 37 (61.67) | 65 (52.00) | 22 (23.66) | 46 (23.59) | <0.001 |
| Yes | 1 (2.04) | 12 (20.00) | 38 (30.40) | 58 (62.37) | 118 (60.51) | |
| Yes, texts only | 1 (2.04) | 10 (16.67) | 22 (17.60) | 13 (13.98) | 31 (15.90) | |
| Missing | | 1 (1.67) | | | | |

**Table 3 Use of health apps before or due to the COVID-19 pandemic (n = 149).**

| Characteristic | Due to COVID-19 Pandemic (n = 81) | Before COVID-19 pandemic (n = 68) | P value |
|---|---|---|---|
| **Age (years)** | | | |
| Mean (SD) | 62.6 (10.4) | 67.8 (10.7) | 0.004 |
| Median (IQR) | 64.0 (12.2) | 69.0 (14.2) | |
| Range | 21.0–82.0 | 32.0–89.0 | |
| **Sex, n (%)** | | | |
| Male | 41 (50.62) | 48 (70.59) | 0.019 |
| Female | 40 (49.38) | 20 (29.41) | |

**Table 4 Future use of a mobile application for chronic disease monitoring by PC teams according to age, sex, daily app usage, and skill (n = 523).**

| Characteristic | Would you use a mobile app to monitor your chronic diseases with the primary care team? | | P value |
|---|---|---|---|
| | No (n = 320) | Yes (n = 203) | |
| **D. you consider yourself skilled in handling apps?, n (%)** | | | |
| No | 212 (66.25) | 44 (21.67) | <0.001 |
| Regular | 69 (21.56) | 57 (28.08) | |
| Yes | 38 (11.88) | 102 (50.25) | |

| Characteristic | Would you use a mobile app to monitor your chronic diseases with the primary care team? | | P value |
| --- | --- | --- | --- |
| | No (n = 320) | Yes (n = 203) | |
| **Do you use apps daily?, n (%)** | | | |
| No | 199 (62.19) | 19 (9.36) | <0.001 |
| Yes | 73 (22.81) | 154 (75.86) | |
| Yes, texts only | 48 (15.00) | 29 (14.29) | |
| **Age (years)** | | | |
| Mean (SD) | 76.6 (10.9) | 66.4 (10.7) | <0.001 |
| Median (IQR) | 79.0 (13.0) | 66.0 (12.0) | |
| Range | 21.0–99.0 | 21.0–95.0 | |
| **Sex, n (%)** | | | |
| Male | 155 (48.44) | 111 (54.68) | 0.200 |
| Female | 165 (51.56) | 92 (45.32) | |

**Table 5 Search for health information on the Internet according to age, sex, education level and skill (n = 523).**

| Variable, n (%) | Do you usually search for health information about treatments, medical conditions, or healthcare on the internet? | | P value |
| --- | --- | --- | --- |
| | No (n = 389) | Yes (n = 134) | |
| **Age (years)** | | | |
| Mean (SD) | 75.0 (11.0) | 65.8 (11.8) | <0.001 |
| Median (IQR) | 77.0 (15.0) | 66.0 (12.2) | |
| Range | 30.0–99.0 | 21.0–91.0 | |
| **Sex, n (%)** | | | |
| Male | 197 (50.64) | 69 (51.49) | 0.920 |
| Female | 192 (49.36) | 65 (48.51) | |
| **Level of studies, n (%)** | | | |
| Primary | 200 (51.41) | 54 (40.30) | <0.001 |
| Secondary or higher | 52 (13.37) | 69 (51.49) | |
| None | 130 (33.42) | 8 (5.97) | |
| **D. you consider yourself skilled in handling apps?, n (%)** | | | |
| No | 242 (62.21) | 14 (10.45) | <0.001 |
| Yes | 61 (15.68) | 79 (58.96) | |

(51.6%, 96/186). Those preferring phone consultations were mainly women (85.7%, 6/7). A higher number of in-person visits was associated with those preferring this modality (t = 3.23; $P < 0.001$).

Among app users, 74.4% (151/203) would use it for chronic disease monitoring, 45.8% (93/203) for treatment follow-up, 66% (134/203) for appointments, 50.7% (103/203) for communication, and 23.1% (47/203) to avoid emergency visits.

### Searching for health information on the internet

Regarding the search for health information on the internet, in Table 5, statistically significant differences were observed showing that younger people (t = 8.05; $P < 0.001$), those with higher education (t = 93.81; $P < 0.001$), and those more skilled in app handling (t = 127.98; $P < 0.001$) searched for health information more frequently.

Lastly, regarding where they usually searched for information, 81.8% (108/134) did so on search engines like Google. Regarding the education level of the participants, those who searched for information on Google or an official channel held secondary or higher education (52.8% (57/108) and 69.2% (9/13), respectively), while those who searched on social media, 71.4% (5/7) had primary education.

## DISCUSSION

### Principal results

This study highlights that we are still not prepared for the widespread implementation of mHealth in nursing consultations for chronic patient care in PC due to the digital divide. Not all patients have equal access to these tools, despite their health benefits.

The primary group of patients in this study consists of older adults with chronic diseases, mostly HTN. This patient profile aligns with data published by the Valencian Community, where 50% of patients visiting a health center are over 65 years old, and HTN is the most prevalent disease (Alòs et al., 2024).

Chronic diseases, including HTN, are increasing in low- and middle-income countries, causing 86% of deaths. The WHO also highlights that advanced age is a vulnerability factor, along with poor diet, a sedentary lifestyle, and smoking (World Health Organization, 2020; World Health Organization, 2023b).

This study identified several variables associated with mobile phone use and online health information-seeking behavior, with age emerging as the most influential factor. These findings are consistent with data from the Spanish National Institute of Statistics (INE), which indicate that both mobile phone use and the search for health-related information decrease significantly with age (Instituto Nacional de Estadística, 2023b). Specifically, 46.1% of individuals aged 65–74 search for health information online, compared to 18.5% of those over 75 and only 7.8% of those over 85.

Furthermore, a discrepancy was observed between the sources consulted by patients and those recommended by healthcare professionals (Monasor-Ortola, Mira-Solves & Esteve-Ríos, 2025). While only 16% of professional recommendations referred to general search engines such as Google, 81.8% of patients reported using them regularly to access health-related information.

In other regions, general online searches among individuals over 60 vary significantly across Latin American countries, depending on their socioeconomic characteristics (Sunkel & Ullmann, 2019). Besides age, the economic profile also influences the use of these technologies. There are large differences between regions, from 92% in the U.S. to 43% in South Asia and 36% in Southern Africa (The World Bank, 2024).

Our findings suggest a more pronounced gender gap in digital engagement than that reported in national statistics. While national data indicate that 96.6% of men and 96% of women have used a mobile phone (*Instituto Nacional de Estadística, 2023b*), our study reveals that men are overrepresented among those with more than 20 years of mobile phone use and are significantly more likely to report digital proficiency. Additionally, men reported greater use of health-related mobile applications. These differences highlight not only disparities in access but also deeper inequalities in digital skills and engagement—possibly linked to traditional caregiving roles more commonly undertaken by women, which may limit their available time and opportunities to engage with technology. Educational level also plays a role: patients with higher education levels searched for health information more frequently (51.5%) and relied more often on official sources. This follows the same INE trend, where 75% of those seeking health information have secondary or higher education (*Instituto Nacional de Estadística, 2023b*).

Regarding diseases, significant differences were found in the use of applications for DM management. This may be due to the increased use of continuous interstitial glucose monitoring systems and glucose control applications. In part, this is because these technologies are covered by the National Health System, allowing free access to these devices for patients with type 1 DM and type 2 DM on intensive insulin therapy. Additionally, their popularity has grown due to the less invasive, less painful, and more accurate self-monitoring of blood glucose levels (*Ministerio de Sanidad, 2022*).

Mobile phone and health app usage depends on multiple factors. However, among patients with chronic diseases, usage increased during the COVID-19 pandemic. Uncertainty, fear, and the need for information drove this trend while the population was in lockdown. During this period, ICTs (Information and Communication Technologies) became the most accessible source of information (*Almalki & Giannicchi, 2021*: *Jansen-Kosterink et al., 2021*). Despite this, only 11.3% of participants in our study used specific health applications during the pandemic. *Gomes-de Almeida, Marabujo & do Carmo-Gonçalves (2021)* confirmed that ICTs, particularly teleconsultation, were well received by DM and HTN patients, although satisfaction decreased with increasing patient age.

In the future, mHealth could primarily be used for clinical variable monitoring and appointment management in chronic patients treated by nursing staff. Additionally, it would likely be better accepted by younger patients and those with greater proficiency in these technologies. Nurses also demonstrate high skills in managing mHealth and believe that these patients would be the primary beneficiaries of this technology (*Mira-Solves et al., 2014*).

These findings are consistent with (*Gamucci Jiménez de Parga et al., 2023*), who studied telemedicine use, such as virtual consultations, where technological competence is crucial for proper utilization. According to *Vidal-Alaball et al. (2023)*, the digital transformation of the healthcare system must address social and digital determinants as factors of inequality. Furthermore, reducing the digital divide is essential to ensure equitable access to ICTs, regardless of patients' sociodemographic characteristics.

To minimize the digital divide, which disproportionately affects older adults and also reflects a gender gap, it would be beneficial to implement targeted interventions, such as digital literacy programs for vulnerable populations (*Vidal-Alaball et al., 2023*). The development of accessible, intuitive, and equitable digital tools is also essential. A well-designed application with these features could improve communication with nursing staff, foster professional–patient interaction, and enhance self-care, particularly among older adults in PC, ultimately contributing to improved perceived health status (*Barón-Miras et al., 2022*). Furthermore, if the goal is to transform and digitalize chronic disease monitoring, emphasis should be placed on training patients in the use of digital technologies and tailoring tools to individual needs through user-friendly interfaces and simplified data entry methods (*Navarro-Martínez, Martinez-Millana & Traver, 2024*). *Sockolow, Buck & Shadmi (2021)* also emphasize that adoption depends on broader contextual factors, such as technological infrastructure and household access, and recommend integrating sensors or voice commands to overcome usability barriers. AI-driven chatbots offer an additional strategy to support interaction through voice-or text-based systems (*Alòs et al., 2024*).

## Limitations

Despite its findings, this study has limitations. No validated instrument was used, as none were found in Spanish, though they exist in English. The questionnaire was administered by a nurse, which could have potentially introduced social desirability and interviewer bias, which a self-administered form could have mitigated. Second, the study excluded patients with chronic cardiovascular diseases treated solely by medical personnel or in private healthcare. Third, socioeconomic status was not considered, even though it is a key variable influencing access to ICTs. Additionally, the survey did not assess whether the use of health-related apps or devices was prescribed by healthcare professionals or self-initiated by the patients, which limits the interpretation of clinical guidance in mHealth use. Lastly, As the study was conducted in a specific Spanish region, findings may not be generalizable. Factors like technological development, household equipment, and healthcare models vary; for instance, in Bismarck-type systems (*e.g.*, Germany, France), family medicine is not the primary healthcare entry point (*Salvador Comino et al., 2016*).

## CONCLUSIONS

This research shows that the patient profile most likely to use mHealth technologies in PC includes younger individuals, men, those with higher educational attainment, and greater digital proficiency. Patients with DM were the most frequent users of these tools for disease management. These findings suggest the presence of potential disparities in mHealth use, which may be indicative of a digital divide; however, such interpretations should be approached cautiously and in light of the study's limitations. The results should be considered applicable to the specific PC context investigated, rather than generalizable to broader populations.

Among those unwilling to use a mobile app for chronic disease monitoring, many preferred in-person consultations. This preference was mainly observed in older adults, emphasizing the need for inclusive strategies in the transition to digital health solutions.

To successfully transform and digitalize chronic disease monitoring, it is essential to promote digital literacy among patients and tailor digital tools to individual needs, through the development of accessible, user-friendly applications with simplified data entry methods. From a global health perspective, chronic disease management cannot be universally standardized, as technological infrastructure and household resources differ significantly across countries and may require substantial investment by either the patient or the healthcare system.

## ABBREVIATIONS

| | |
|---|---|
| **AI** | Artificial intelligence |
| **App** | Application |
| **BHZ** | Basic health zone |
| **BMI** | Body Mass Index |
| **CHF** | Congestive heart failure |
| **COPD** | Chronic Obstructive Pulmonary Disease |
| **DBP** | Diastolic Blood Pressure |
| **DHD** | Denia Health Department |
| **DM** | Diabetes Mellitus |
| **EC** | European Commission |
| **HTN** | Hypertension |
| **ICT** | Information and communication technologies |
| **INE** | National Statistics Institute of Spain |
| **IoMT** | Internet of Medical Things |
| **mHealth** | Mobile Health |
| **PC** | Primary care |
| **SBP** | Systolic Blood Pressure |
| **VC** | Valencian Community (Spain) |
| **WHO** | World Health Organization |

## ACKNOWLEDGEMENTS

The authors thank the nursing staff of the 11 PC zones of the DHD, especially the Benissa Health Center, for their support in data collection. They also acknowledge the use of ChatGPT-4, employed solely to assist with English writing.

### Funding

The authors received no funding for this work.

## Competing Interests

The authors declare that they have no competing interests.

## Author Contributions

- Daniel Monasor Ortola conceived and designed the experiments, performed the experiments, analyzed the data, prepared figures and/or tables, authored or reviewed drafts of the article, and approved the final draft.
- José Joaquín Mira conceived and designed the experiments, analyzed the data, prepared figures and/or tables, authored or reviewed drafts of the article, and approved the final draft.
- Antonio Esteve Ríos conceived and designed the experiments, analyzed the data, prepared figures and/or tables, authored or reviewed drafts of the article, and approved the final draft.
- Virginia García Ferrer performed the experiments, analyzed the data, authored or reviewed drafts of the article, and approved the final draft.

## Human Ethics

The following information was supplied relating to ethical approvals (*i.e.*, approving body and any reference numbers):

The study was approved on 7 November 2022 by the DHD Research Commission, which oversees ethical compliance

## Ethics

The following information was supplied relating to ethical approvals (*i.e.*, approving body and any reference numbers):

The study was approved on 7 November 2022 by the DHD Research Commission, which oversees ethical compliance

## Data Availability

The raw data is available in the Supplemental File.

## Supplemental Information

Supplemental information for this article can be found online at http://dx.doi.org/10.7717/peerj.20130#supplemental-information.

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
