# Peer review of "Exploring the use of mobile health among patients with cardiometabolic and respiratory chronic diseases in primary care nursing: a cross-sectional study"

_PeerJ, doi:10.7717/peerj.20130_

## Round 0.1 · original submission · Major Revisions

**Language Note:** The review process has identified that the English language must be improved. PeerJ can provide language editing services - please contact us at [email protected] for pricing (be sure to provide your manuscript number and title). Alternatively, you should make your own arrangements to improve the language quality and provide details in your response letter. – PeerJ Staff

·

Basic reporting

This paper presents results of an observational study concerning use of mHealth apps in management of cardiovascular and respiratory diseases by primary care nursing.

/* Clear and unambiguous, professional English used throughout body text - OK */

/* Literature reference require revision - work needed */
1. Line 77 - Introduction - “The "Seville Declaration ...”, resulting from Spain's national conference on chronic diseases” - This reviewer understands this declaration to be over a decade old. The authors may wish to cite this as a long-standing aspirational statement, but this reviewer suggests incorporating more recent position statements e.g.:

https://www.who.int/publications/i/item/9789240020924
https://www.who.int/publications/m/item/global-initiative-on-digital-health

It may also be warranted to acknowledge EU work in this space:
https://health.ec.europa.eu/ehealth-digital-health-and-care/digital-health-and-care_en

2. Line 85 - Two further taxonomies by which mHealth apps have been defined may be of use to the authors here when defining types of mHealth apps - (a) First they can be simply classified as either passive (static health info pages, symptom diaries, manual data entry into an app) or active (using device sensors to capture health information). see:

Herron, J. (2016). Bad Apps: mHealth Apps Doubling as Medical Devices. Journal of Electronic Resources in Medical Libraries, 13(4), 177–181. https://doi.org/10.1080/15424065.2016.1256800

(b) mHealth apps can also be classified based on criticality (information, primary prevention, secondary and tertiary prevention and Data Analysis app types) -

see table 2 on p 15 of:
https://www.has-sante.fr/upload/docs/application/pdf/2017-03/dir1/good_practice_guidelines_on_health_apps_and_smart_devices_mobile_health_or_mhealth.pdf

3. Line 101 - Introduction - As with medical staff, there are challenges to finding evidence-based apps for prescription to patients which are proven to be safe and effective (with potential medico-legal ramifications arising from prescribing or suggesting use of a ‘bad app’). The authors might also consider this as a barrier to wider adoption of mHealth prescription by nurses to patients.

4. Line 101 - Introduction - Readers would benefit from consideration of the roles of nursing staff in supporting mHealth use e.g training patients in app use, coaching and encouraging compliance with app, assisting in capture of data from apps and assimilation into patients medical record, patient monitoring and escalation of any adverse reactions/finding arising from app use.

Minor items
* * *
5. Line 27 - Abstract - Background - “Chronic diseases, including cardiometabolic and respiratory conditions, are major contributors to global mortality, placing a significant burden on primary care systems, with high costs and increased dependency” - (a) Could the authors consider revising, updating and simplifying this complex opening sentence. (b) In addition to “global mortally”, this reviewer suggests that morbidity contributes significantly to burdens on health care systems (as acknowledged later in the Introduction) and (c) This reviewer is unsure as to the meaning of “and increased dependency” - ? is this increasing dependancy on primary health case systems.

6. Line 27 - Abstract - Background - “Digital tools such as mHealth have emerged as key resources for optimizing chronic disease management in primary care” This reviewer believes that surgical and/or pharmacological interventions are key resources, and that emergent digital tools such as mHealth show promise as adjuncts in management of some chronic diseases in primary care.

7. Line 30 - Abstract - Background - Despite their potential, the adoption of mHealth by nurses in chronic care remains limited, with few studies focusing on its use for patients with multiple chronic conditions. - This reviewer recommends that in this sentence (and also in the title of this article), the authors make it clear as to the scope and type of of mHealth apps considered here. It is unclear whether this article pertains to selection and use of mHealth apps for use by nursing staff themselves for the purposes of supporting and delivering care to patients. In the next sentence (Line 32), the authors state that “This study aims to evaluate the utilization and implementation of mHealth among patients ... who are under the care of primary care nursing services”. It is important for readers to understand whether the mHealth apps in question in this study were prescribed by health professionals to patients as part of evidence-based treatment regimens, were the gaps self-selected by patients or whether nursing staff suggested or recommended use of given apps to patients.

/* Self-contained with relevant results to hypotheses - OK */

Experimental design

/* Original primary research within Aims and Scope of the journal - OK */

/* Research question well defined, relevant & meaningful. - work needed */

8. Line 101 - Introduction - “However, nurses do not usually recommend its use to chronic patients ” and Line 105 “Few studies have analyzed the application of mHealth by nurses in patients with patients with multiple chronic conditions” - in many countries, mHealth apps fall within the remit of medical device registration and legislation. As mentioned by this reviewer in earlier comments regarding the Abstract - Background section, this reviewer feels that the authors need to made clear as to whether nursing staff recommend (i.e prescribe) use of particular apps to patients?

As with medical staff, there are challenges to finding evidence-based apps for prescription to patients which are proven to be safe and effective (with potential medico-legal ramifications arising from prescribing or suggesting use of a ‘bad app’). The authors might also consider this as a barrier to wider adoption of mHealth prescription by nurses to patients.

9. Line 112 - “Thus, the objective of this study was to describe the frequency of use and application of mHealth technologies in patients with chronic cardiometabolic and/or respiratory diseases followed by nursing staff in PC” - could the authors consider the wording in this sentence and align it with the statement in the Abstract - background - Line 24 - “... who are under the care of primary care nursing services”
*/
/* Rigorous investigation performed to a high technical & ethical standard - OK */

/* Methods described with sufficient detail & information to replicate - work needed */

10. Line 123 -Materials and Methods - Participants - “The study focused on residents of Marina Alta, VC, Spain, within the DHD, part of the VC Health Ministry” - for the benefit of readers unfamiliar with this region, do residents receive free health care or do they pay as private patients?

11. Line 129 - Materials and Methods - Participants - If the authors agree that a purposive sampling methodology was employed, could this be stated as such.

12. Line 129 -Materials and Methods - Participants - “Participants were consecutively recruited by primary care nurses throughout the study period, reaching 200% of the initially estimated minimum sample size” To help readers better understand the population from which this study cohort is drawn, could the authors describe the following:

(a) what was the sequence of presentation for prospective study participants i.e. medical practitioner consultation followed by referral to primary care nursing care? Line 130 indicates that “...patients enrolled in a chronic nursing program in PC but didn’t specify whether they were first timers or returning patients”

(b) Was the study protocol implemented at one or more sites, and run one or more nursing staff members (if multiple, how was consistency in approach/invitation to participate maintained?)

(c) Could the authors indicate whether participants received any remuneration or reward for taking part in this study.

Validity of the findings

/* Impact and novelty not assessed. Meaningful replication encouraged where rationale & benefit to literature is clearly stated - work needed */

13. Line 184 Results - “Statistically significant differences were found, indicating that mobile device use was higher among younger individuals and males (P<.001)” - for the benefit of readers, could the authors updated instances in the Results section where P values are quoted to display as (test statistic=1.23; p<0.01)

14. Line 188 - Use of Mobile Health applications and Line 204 - Use of Mobile Apps for Chronic Disease Monitoring in PC - This reviewer was unable to find results regarding mHealth adoption and use by patients in this cohort (as pertaining to disease management by primary care nursing. How many were prescribed use of an mHealth app by a doctor, nurse, pharmacist, nutritionalist or other health professional at/before the time of their presentation to the nursing clinic? (first time attendees might have self-selected apps compared with those who are under longer term primary care nursing?)

15. Line 204 - Use of Mobile Apps for Chronic Disease Monitoring in PC - “In-person consultation was preferred by 58.3% (186/319), predominantly older adults (mean age 74; SD=11.9; P<.001) and men (51.6%, 96/186). Those preferring phone consultations were mainly women (85.7%, 6/7). A higher number of in-person visits was associated with those preferring this modality (P<.001)” - could the authors clarify the meaning of “preferred” here - is this a preference for consultations per se` or specifically concerning mHealth.

16. Line 309 - Limitations - Could the authors reflect on earlier comments from this reviewer regarding prescribed vs self-prescribed apps - future study design might incorporate assessment of prescribed mHeath.

/* All underlying data have been provided; they are robust, statistically sound, & control - OK/

/* Conclusions are well stated, linked to original research question & limited to supporting results - Work needed */

17. Line 321 - Conclusions - Could the authors relocate any sentences currently in the Conclusion section which contain citations and move them into the Discussion section. Readers expect that the Conclusion briefly states what has been done, what the results were, and what it all means.

18. Line 323 - Conclusions - "This research shows that the patient profile that most uses mHealth in PC is younger, male, has a higher level of education, and greater proficiency in handling ICTs, with patients with DM making the most use of these tools for managing their condition. This identifies a digital divide that creates inequalities in access to this technology. " These statements need to be be tempered by the study design used and the purposive (non-probaility) cohort investigated, and should be qualified as pertaining the single primary care setting investigated.

·

Basic reporting

Dear Authors
Please find my comments on your manuscript below. This study explores the use of mHealth by primary care nurses for cardiometabolic and respiratory cases. The study’s results identified that Patients with diabetes were the primary users of mHealth, and age was the predominant factor influencing the mHealth management. Most importantly, you highlighted the digital divide as a main study result.
I will be commenting on the manuscript section.
Title:
Short, concise, but misleading; the “primary care nursing” does not refer to the study population, “patients”. Study design should be identified in the title.
Suggest rephrasing to: Exploring the use of mHealth among patients with cardiometabolic and respiratory diseases in primary care nursing: a cross-sectional study
Abstract:
• Mention the sampling technique you followed under the methods section.
• Mention how the data was analyzed under the methods section. the software and the statistical tests applied
Introduction:
• Line 64: Write PC in full on the first mention.
• The flow and coherence of the introduction are violated. I suggest collating and summarizing paragraphs in a rational flow rather than writing fragmented ideas and sentences.
• Line 105: delete the repetition “patients with”
• Lines 112-114: rewrite the objective clearly; your main objective was not exploring the frequency; you explored the overall usage of mHealth among the mentioned patient group
• The main focus of the study results and conclusion is the digital divide, but where is this defined in the introduction? I suggest adding a paragraph identifying the concept and the aspects of the digital divide, and how it can be evaluated.
• I suggest adding a clear statement on whether the ad hoc questionnaire is valid for evaluating the digital divide or not.

Experimental design

Methods:
• What is the sampling technique?
• Justify the use of an ad hoc questionnaire instead of a validated survey.
• You mentioned that the nurses collected the research data? Are they trained to commit to this role? What is the role of researchers in this case?
• What are the measures taken to ensure the validity and reliability of the ad hoc questionnaire? Is it valid to evaluate the mHealth usage among study participants?
• What is the rationale for asking a question related to using Health apps during COVID-19, which was five years ago? Have you clearly defined the health apps versus commercial apps to the study participants to avoid confusion?

Validity of the findings

Results:
• Wrong interpretation of study findings; patients with diabetes were the primary users of mHealth doesn’t indicate a digital divide. Given that authors have not identified what the digital divide is or how it can be evaluated.

Discussion
• Lines 263-256: 96.6% of men have used a mobile phone compared to 96% of women … the authors implied that these results may indicate a gender gap. The provided findings are insufficient to decide the gender gap.
Conclusion
• Lines 223-226: The study findings regarding age, education, and diagnosis are insufficient to determine the existence of a digital divide.

Additional comments

- This work is non-coherent and lacks novelty.
- The applied methods are not rigorous. The selection of the ad hoc questionnaire lacks scientific merit, making it neither valid nor reliable for addressing the research question.
- The study findings don’t support the mentioned conclusion related to the digital divide; wrong interpretation.
- What are the implications of this study?
- The manuscript should be revised to improve its flow and enhance coherence, especially the introduction. It should also be submitted for language and grammar editing.

---

## Round 0.2 · accepted · Accept

Thank you for revising your manuscript to address the reviewers' concerns. Reviewer 1 now recommends acceptance and I am satisfied that the comments of Reviewer 2 have been addressed. The manuscript is now ready for publication.

·

Basic reporting

Improvements in readability and adoption of suggestions made by this reviewer are noted in this updated manuscript, with particular reference to the Abstract and Introduction sections. The paper was self-contained, with relevant results reported for hypotheses.

Where suggested changes were not made or alternative changes were applied, the authors have provided adequate justification.

Literature references were sufficient to provide background/context.

The article structure was appropriate, with raw data shared.

Experimental design

Original primary research was within the Aims and Scope of the journal.

Research question well defined, relevant & meaningful, namely examine the overall use and implementation of mHealth technologies among patients with chronic cardiometabolic and/or respiratory diseases who are under the care of primary care nursing services.

Research was performed to a high technical & ethical standard, with sufficient detail of methods to allow others to replicate.

Validity of the findings

Impact and novelty were identified.

Underlying data was provided. Statistical methods were robust.

Conclusions are well stated and linked to the original research question.

Additional comments

This reviewer congratulates the authors on this revised manuscript.